



# Mercury emissions of a coal fired power plant in Germany

**Andreas Weigelt[1,*], Franz Slemr[2], Ralf Ebinghaus[1], Nicola Pirrone[3], Johannes**
**Bieser[1,4], Jan Bödewadt[1], Giulio Esposito[3], Peter F.J. van Velthoven[5]**
[1]Helmholtz-Zentrum Geesthacht (HZG), Institute of Coastal Research, Geesthacht, Germany
[2]Max-Planck-Institute for Chemistry (MPI-C), Department of Atmospheric Chemistry,
Mainz, Germany
[3]National Research Council (CNR), Institute of Atmospheric Pollution Research, Rende, Italy
[4]Deutsches Zentrum für Luft- und Raumfahrt (DLR), Institute of Atmospheric Physics,
Oberpfaffenhofen, Germany
[5]Royal Netherlands Meteorological Institute (KNMI), Chemistry and Climate Division, De
Bilt, Netherlands
*now at: Federal Maritime and Hydrographic Agency (BSH), Hamburg, Germany
Correspondence to: A.Weigelt (Andreas.Weigelt@bsh.de), F. Slemr (franz.slemr@mpic.de)
andreas.weigelt@bsh.de
franz.slemr@mpic.de
ralf.ebinghaus@hzg.de
pirrone@iia.cnr.it
johannes.Bieser@hzg.de
jan.boedewadt@hzg.de
esposito@iia.cnr.it
velthove@knmi.nl



**Abstract**
$Hg/SO_2$, $Hg/CO$, $NOx/SO_2$ emission ratios (ERs) in the plume of coal fired power plant
(CFPP) Lippendorf near Leipzig in Germany were determined within the European
Tropospheric Mercury Experiment (ETMEP) aircraft campaign in August 2013. GOM
fraction of mercury emissions was also assessed. Measured $Hg/SO_2$ and $Hg/CO$ ERs were
within the measurement uncertainties consistent with the ratios calculated from annual
emissions in 2013 reported by the CFPP operator, the $NOx/SO_2$ ER was somewhat lower.
GOM fraction of total mercury emissions, estimated by three independent methods, was
~10% with an upper limit of ~25%. This result is consistent with findings by others and
suggests that GOM fractions of ~40% of CFPP mercury emissions in current emission
inventories are overestimated.
**1 Introduction**
Mercury and especially methyl mercury which bio-accumulates in the aquatic nutritional
chain are harmful to humans and animals (e.g. Mergler et al., 2007; Scheuhammer et al.,
2007; Selin, 2009; and references therein). Therefore, its emissions are on the priority list of
several international agreements and conventions dealing with environmental protection and
human health, including the United Nations Environment Program (UNEP) Minamata
convention on mercury (www.mercuryconvention.org). Mercury is emitted to the atmosphere
from a variety of natural (e.g. volcanic activity, evaporation from ocean and lakes) and
anthropogenic sources (e.g. coal and oil combustion) (Mason et al., 2009; Pirrone et al.,
2010). Coal-fired power plants (CFPPs) are believed to account for most ($\geq 56\%$) of mercury
emitted by stationary combustion sources which constitute 35 – 77% of all anthropogenic
emissions (Pirrone et al, 2010; Chen et al., 2014; Ambrose et al., 2015).
Mercury from CFPPs is emitted as gaseous elemental mercury (GEM), gaseous oxidized
mercury (GOM) and particulate bound mercury (PBM). Elemental mercury has a high vapour
pressure, is virtually insoluble in water resulting in a long residence time in the atmosphere of
about 1 yr (Selin, 2009). GOM with its high solubility and low vapour pressure is readily
washed and rained out as are the particles carrying particulate mercury (PM). GOM and PM
are believed to be in equilibrium (Rutter and Schauer, 2007; Amos et al., 2012). GOM is thus
a major driver for the global mercury deposition and is estimated to make up more than 50%
of the total Hg deposition (Zhang et al., 2012a; Bieser et al., 2014).



There are only two sources of GOM in the atmosphere: primary GOM emissions from
anthropogenic sources and the oxidation of elemental mercury. The major anthropogenic
mercury sources on a global scale are small scale artisanal gold mining (SSAG) and coal
combustion (Pirrone et al. 2010). While SSAG emits solely elemental mercury, the CFPP
emissions in emission inventories are estimated to have a GOM fraction between 35% and
40% (Pacyna et al., 2006; Wilson et al., 2010; EPA, 2011). However, global and regional
model studies have repeatedly indicated that models are overestimating atmospheric GOM
concentrations (Zhang et al., 2012b; Kos et al., 2013; Bieser et al., 2014). Possible
explanations for this are an overestimation of the GEM oxidation rates or the overestimation
of the amount of GOM emitted by CFPPs. The latter has been hypothesized to be due to a fast
reduction of GOM inside the plume (Zhang et al., 2012b; Kos et al., 2013).
While the operators of CFPPs are forced to measure and report the amount of mercury
released into the atmosphere, there is only little knowledge on the speciation of these
emission sources. That is because of varying composition of burnt coal, complex chemistry in
the stack gases (e.g. Schofield, 2008; Ernest Tatum et al, 2014) and the large number of
different methods used to clean CFPP flue gases with very different percentage of gaseous
oxidized mercury (GOM) to total mercury ranging from less than 10% up to 90% (Wang et
al., 2010, Schuetze et al., 2012, and references therein). Analytical problems also contribute to
the uncertainty: the current emission monitoring systems are not sensitive enough to measure
and speciate low mercury concentrations in flue gases of modern CFPPs (Mayer et al., 2014).
Moreover, there has been evidence that the current ambient air measurement systems might
not capture all oxidized mercury species with similar efficiency (Jaffe et al., 2014; Gustin et
al., 2015a, Weiss-Penzias et al., 2015).
The European Tropospheric Mercury Experiment (ETMEP) was carried out in July/August
2012 (ETMEP-1) and August 2013 (ETMEP-2) to measure local emissions, vertical profile
from inside the boundary layer to the lower free troposphere, and horizontal distribution of
mercury over Europe. In total 10 measurement flights were performed over Italy, Slovenia,
and Germany with two propeller aircraft. The ETMEP-1 campaign focused on volcanic
emissions of Etna. The objectives of the ETMEP-2 campaign were a) to obtain vertical
mercury profiles above several sites in central and southern Europe (Weigelt et al., 2016), b)
to assess horizontal distribution of mercury concentrations during the flight from Italy to



Germany, and c) to determine mercury emission ratios for a coal-fired power plant (CFPP)
near Leipzig. Here, we present the measurements of CFPP emissions and their speciation.
**2 Experimental**
The power plant under investigation is located in Lippendorf, a small village ca 15 km south
of Leipzig. The CFPP of Lippendorf consists of two units with 934 MW gross power each. It
has been in operation since 2000 and belongs with a net efficiency of 42.6% to one of the
most modern and efficient CFPPs in Europe. About 10 million metric tons of brown coal with
rather high sulphur content from a nearby open pit mine are burnt annually. The $SO_2$
emissions are reduced by flue gas desulfurization (FGD) system using wet washing with CaO
suspension. Despite the efficient FGD cleaning, the CFPP of Lippendorf ranks 4[th] most
harmful emitter in Germany (Preiss et al., 2013) and 14[th] most harmful emitter in Europe
according to the European Environment Agency (EEA, 2011) with respect to health. Annual
emissions reported by the operator of the CFPP Lippendorf for 2013, the year of our
measurements, were: $1.18*10^{13}$ g $CO_2$, $1.21*10^{10}$ g $SO_2$, $7.91*10^9$ g NOx, $7.55*10^8$ g CO, and
$4.1*10^5$ g Hg, among other pollutants. Mercury limit emission values (LEVs) of large
combustion plants in Germany are stipulated by ordinance (Federal Law) from 2004 and its
revision in 2013 to 50 µg m$^{-3}$ as a half hour average, 30 µg m$^{-3}$ as a daily average, and 10 µg
m$^{-3}$ as an annual average (Mayer et al., 2014). Continuous monitoring of mercury emissions is
mandatory but only annual total (unspeciated) mercury emissions have to be reported.
The measurement campaign described above was performed with a CASA 212 two engine
turboprop aircraft (Fig. 1a) operated by Compagnia Generale Ripreseaeree
(http://www.terraitaly.it/). The CASA 212 with a maximum payload of 2.7 tons can carry the
measurement instruments, different service instruments, the power supply, two pilots, and 5
operators. With a normal cruising speed of ~ 260 km h$^{-1}$ its range is ~ 1600 km. Although the
maximum flight level of the unpressurized aircraft is 8500 m, the maximum altitude of
ETMEP-2 flights without oxygen supply was limited to ~3000 m above sea level (a.s.l.),
The aircraft was equipped with a gas inlet system (Fig. 1b) which had been developed and
manufactured at the Helmholtz-Zentrum Geesthacht. The gas inlet was designed for the
cruising speed of the CASA 212 of ~ 72 m s$^{-1}$. A diffuser tube reduced the air speed to ~ 5 m
s$^{-1}$. About 120 l min$^{-1}$ (ambient conditions) enters the inlet at the cruising speed of 260 km h$^{-1}$.
The air sample is taken in the centre of the diffuser tube with a flow rate of ~ 25 l min$^{-1}$. The





remaining flow of 95 l min$^{-1}$ is directed to the back of the inlet where the air speed is
increased by a nozzle and the air exits. By replacing the inlet and outlet nozzle with smaller or
larger ones, this inlet system can be fitted to other aircraft with a different cruising speed. In
the expanded area (behind the main sample line) the air temperature (T), static pressure (p),
and relative humidity (rH) are measured. To avoid adsorption losses of sticky trace gases, the
internal surface of the inlet system was coated with Teflon and only PFA tubing was used for
the sampling lines. The outside of the inlet was coated with copper to avoid electrostatic
charging. The inlet was fastened onto a 90 cm long telescope tube (6 cm diameter) which was
mounted in a hole on the floor fuselage via a sliding guide. After take-off, the tube was
pushed down by ~40 cm from inside the aircraft, to ensure that the inlet nozzle is outside the
aircraft boundary layer. Before landing the tube was pulled back into the aircraft to protect it
from damage by objects whirled up by the front wheel. The inlet and the telescope tube were
equipped with heaters to prevent icing but during the ETMEP measurements the heating was
always switched off because the measurement flights were carried out in summer at altitudes
below 3000 m a.s.l. The tubing from the inlet to instruments (~2.5 m long 3/8'' main sample
tube with PFA manifolds to instruments) was not heated. The temperature inside the cabin
was 18 to 30°C.
The aircraft was equipped with three mercury measurement instruments: a Lumex
RA-915AM, a Tekran 2537B, and a Tekran 2537X (cf. Tab. 1). The Lumex RA-915 AM is
based on atomic absorption spectroscopy (AAS) with Zeeman background correction
(Sholupov et al., 2004) and as such measures specifically only gaseous elemental mercury
(GEM) with a temporal resolution of 1 s. Its raw signal is noisy (about $\pm$ 4 ng m$^{-3}$ with a
temporal resolution of 1 s) and is dependent on pressure and temperature. Nevertheless, the
fast response of the instrument is very useful to detect GEM in rather narrow highly
concentrated plumes at a cruising speed of about 72 m s$^{-1}$. Because of thermal drifts its zero
was measured every 4 min for 1 min.
The Tekran 2537B and 2537X analysers are based on preconcentration of mercury and its
compounds on gold traps (Slemr et al., 1979), thermodesorption, and detection by cold vapour
atomic fluorescence spectroscopy (CVAFS). Although CVAFS can detect only GEM,
mercury compounds are converted to GEM during adsorption or thermodesorption (Slemr et
al., 1978) and, consequently Tekran instruments measure total gaseous mercury (TGM). The
instruments use two gold traps to ensure a continuous measurement: while one is adsorbing





mercury during sampling, the other one is being analysed and vice versa. The highest
temporal resolution of the Tekran instruments of 150 s is given by the time necessary for the
thermodesorption of mercury from the gold traps and their cooling. The Tekran 2527X
analyser was run with quartz wool trap upstream of the instrument, which removes gaseous
oxidized mercury (GOM) and aerosol particles with particle bound mercury (PBM) but no
GEM from the air stream (Lyman and Jaffe, 2011; Ambrose et al., 2013). The Tekran 2537B
analyser was operated as backup instrument without a quartz wool trap.  The Teflon made
(PFA and PTFE) aircraft gas inlet and tubing system are similar to the CARIBIC trace gas
inlet for which high GOM transmission was qualitatively demonstrated. Based on the short
residence time (0.3 sec) in the tubing to the instrument, the conditions as during an
international field intercomparison (Ebinghaus et al., 1999), and higher GOM concentrations
in the plume than in ambient air, we presume Tekran measurements without quartz wool trap
represent total gaseous mercury (TGM = GEM + GOM). Therefore, the Tekran 2537B
measurement are believed to represent TGM concentrations whereas those by Tekran 2537X
GEM concentrations, both with an uncertainty of 12.5%. The uncertainty has been calculated
by Weigelt et al. (2013) using two different approaches according to ISO 20988 type A6 and
ISO 20988 Type A2. This uncertainty complies with the quality objective of the EU air
quality directive 2004/107/EC. The instrumental setup in the aircraft was almost identical and,
therefore, we expect the uncertainty to be very similar.
Direct estimation of the GOM concentrations was made using three manual KCl denuder
samples taken during the vertical profiles (sampling time 1 hour or longer, sampling flow rate
6.4 l/min at standard temperature and pressure (STP; T=273.15 K, p=1013.25 hPa),
corresponding to ~ 10 l/min at ambient temperature and pressure in 3000 m a.s.l and
controlled using a mass flow controller): one downwind of the Lippendorf CFPP, one upwind
over the city of Leipzig (both on August 21, 2013), and one over the GMOS master site
"Waldorf" in northern Germany on August 22. Two blank samples were also taken by KCl
denuders handled exactly in the same way as the samples (denuder preparation, installation to
sampling setup, storage, analysis) but without sucking sample air through them. After all
flights had been finished, the KCl denuders were analysed for their total GOM loads in the
laboratory. Despite a relatively high uncertainty of about $\pm$ 5 pg m$^{-3}$, the method provides
semi-quantitative information about GOM concentration in the plume.



We note that both methods used here to estimate GOM concentrations are subject to
interferences. GOM captured by quartz wool can be released by higher air humidity (Ambrose
et al., 2015) and KCl traps and denuders can release GOM in presence of high ozone and
water concentrations (Lyman et al., 2010; Huang and Gustin, 2015). These interferences may
result in overestimation of GEM and underestimation of GOM emissions. GEM measured by
Lumex is not subject to any known interference.
For the identification and characterization of different air masses carbon monoxide (CO),
ozone ($O_3$), sulphur dioxide ($SO_2$), nitric oxide (NO), nitric dioxide ($NO_2$), and the basic
meteorological parameters temperature (T), pressure (p), and relative humidity (rH) were
measured simultaneously with high temporal resolution. Instrument details including the
estimated measurement uncertainty are summarised in Table 1. Uncertainties were calculated
according to the individual instrument uncertainty given by the manufacturer and the
calibration gas accuracy (CO, $O_3$, $SO_2$, NO). All instruments were protected from aerosols
using PTFE filters (0.2 µm pore size). Model meteorological data like potential vorticity,
equivalent potential temperature, relative and specific humidity, cloud cover, cloud water
content, three-dimensional wind vector, as well as five day backward trajectories were
calculated every 150 s along the aircraft flight tracks for additional information. These
calculations are based on meteorological analysis data from the European Centre for Medium-
Range Weather Forecasts (ECMWF) and the TRAJKS trajectory model (Scheele et al., 1996).
Before take-off all instruments were warmed up for at least 45 minutes, using an external
ground power supply. During the starting of the engines the power was interrupted for less
than 3 minutes. Since 45 minutes were too short to stabilize the Tekran 2537 internal
permeation source, these instruments were calibrated only after each measurement flight
before the engine shut down. All data were recalculated, using the post flight calibration. The
pressure in the fluorescent cells of both Tekran instruments was kept constant using upstream
pressure controllers at the exits of the cells. This eliminated the known pressure dependence
of the response signal (Ebinghaus and Slemr, 2000; Talbot et al., 2007). The Lumex analyser
has a much shorter warm up time of less than 10 minutes and was, therefore, calibrated before
take-off with the internal calibration cell. The CO instrument calibration takes 60 seconds and
was, therefore, performed during the measurement flights every 20 minutes with external
calibration gas. The $O_3$, $SO_2$, $NO/NO_2$ instruments have a fairly constant signal response and
were thus calibrated before and after the ETMEP-2 measurement campaign. Multipoint $SO_2$
and NOx calibration was made using dilution (Environics 300E calibrator) of certified
standard gases. $NO_2$ conversion efficiency was determined using gas phase titration. The
factory calibration was used for the pressure, temperature and relative humidity sensors. The
measurements were synchronized using their individual delay and response times. Please note
that all mercury (TGM, GEM, and GOM) concentrations are reported at standard temperature
and pressure (STP; T = 273.15K, p = 1013.25 hPa). At these standard conditions 1 ng m$^{-3}$
corresponds to a mixing ratio of 112 ppqv (parts per quadrillion by volume).

## 3 Vertical distribution and Hg/SO$_2$, Hg/CO, NOx/SO$_2$ emission ratios

The measurements were carried out on August 21 and 22, 2013. On August 21 between 9:30
and 11:20 UTC the aircraft flew many circles at different altitudes downwind of a CFPP
Lippendorf (51°11`N, 12°22`E) followed between 11:25 and 12:20 UTC by a vertical profile
upwind of CFPP Lippendorf over the city centre of Leipzig (51.353°N, 12.434 °E). Between
8:30 and 10:00 UTC of August 22 another vertical profile above the GMOS master site
"Waldhof" (52°48`N, 10°45`E, about 200 km from Leipzig on the line connecting Leipzig and
Hamburg) was flown, followed between 10:00 and 10:35 UTC by additional measurements
downwind of the CFPP Lippendorf. Each vertical profile consists of at least seven horizontal
flight legs, consisting of circles and altogether lasting 5 - 10 minutes each. The flight legs
started inside the boundary layer at about 400 m above ground and ended at 3000 m a.s.l. The
tracks of the flight in the CFPP plume on August 21 and August 22 are shown in Figure 2a
and 2b, respectively. The CFPP plume was encountered in the distance of ~ 7.5 km from the
plant at an altitude of 1900 m a.s.l. on August 21 and in the distance of ~ 5 km at 1500 – 1650
m a.s.l. on August 22. With a horizontal wind speed of 2.4 and 1.5 m s$^{-1}$ on August 21 and 22,
respectively, the age of the plume was ~0.9 h on both days.
Figures 3 and 4 show data from the flight sections with CFPP plume encounters on August 21
and 22, 2013, respectively. The plume encounters lasted 1 – 2 min and are clearly indicated by
elevated SO$_2$, NOx (NOx = NO + NO$_2$), and GEM concentrations measured by Lumex. CO
and rH enhancements are hardly visible on August 21 but are clearly recognizable on August
22. Tekran instruments with a temporal resolution of 150 s are too slow to resolve individual
plume encounters but they also show a broad peak of enhanced GEM (Tekran 1 with quartz
wool trap) or TGM (Tekran 2) concentrations. The difference between TGM measured by
Tekran 2 and GEM measured by Tekran 1 is small (on average 0.087 ± 0.117 ng m$^{-3}$ (n = 8)



on August 21 and $0.063 \pm 0.079$ ng m$^{-3}$ (n = 12) on August 22) and varies between -0.064 and
+0.354 ng m$^3$ on both days. The average differences are not significantly different from zero
and neither do the maximum and minimum differences exceed the combined uncertainty of
the difference of 17.7%. On August 21 the plume was encountered several times at an altitude
between 1600 and 2500 m a.s.l. The most pronounced encounters numbered 1 – 4 were found
at an altitude of 1800 – 2250 m a.s.l. On August 22 the plume was encountered 3 times at a
flight level of 1550 m and 3 times at 1650 m a.s.l. The numbered plume encounters were
selected for quantitative evaluation.
Figure 5 shows the vertical distribution of the values measured downwind the Lippendorf
CFPP. The vertical profiles above Leipzig and Waldhof are discussed together with further
profiles in Weigelt et al. (2016). In Figure 5 the squares represent the constant flight level
measurement points (2 measurements with 2.5 minutes each). The stars represent the
measurements when climbing between two flight levels (2.5 min average). Therefore the data,
indicated as squares are more significant and the data illustrated as stars do provide additional
information on the vertical structure. Please note that the rH, air temperature (T), and the
potential temperature ($\theta$) are plotted with high temporal resolution (1 s) in the rightmost
panel. The rH can be used to distinguish between boundary layer- and free tropospheric air.
Usually inside the planetary boundary layer (PBL) the relative humidity is much higher than
in the free troposphere (Spencer and Braswell, 1996).
The lower four horizontal flight legs (570 to 1340 m a.s.l.) show typical northern hemispheric
GEM and TGM background concentration of ~1.6 ng m$^{-3}$ without any vertical gradient. CO,
$O_3$, $SO_2$, as well as NO and $NO_2$ also show no vertical gradient, indicating a well-mixed PBL.
This is in agreement to the other vertical profiles measured during ETMEP-2 campaign
(Weigelt et al., 2016). From the fifth flight leg (1630 m a.s.l.) upward the GEM and TGM
concentration increases towards the PBL top (Tekran 1 (GEM): 1.7 ng m$^{-3}$ at 1630 m a.s.l.;
2.6 ng m$^{-3}$ at 1940 m a.s.l.; Tekran 2 (TGM): 1.7 ng m$^{-3}$ at 1630 m a.s.l.; 2.8 ng m$^{-3}$ at
1940 m a.s.l.; Lumex (GEM): 2.1 ng m$^{-3}$ at 1630 m a.s.l.; 2.4 ng m$^{-3}$ at 1940 m a.s.l.). The
increasing concentration is also captured by the measurements during the flight level change
(Tekran 1 (GEM): 1.7 ng m$^{-3}$ at 1540 m a.s.l.; 2.1 ng m$^{-3}$ at 1800 m a.s.l.; Tekran 2 (TGM):
1.7 ng m$^{-3}$ at 1540 m a.s.l.; 2.3 ng m$^{-3}$ at 1800 m a.s.l.; Lumex (GEM): 1.8 ng m$^{-3}$ at 1540 m
a.s.l.; 2.2 ng m$^{-3}$ at 1800 m a.s.l.; stars in Fig. 5). As indicated by the abrupt decrease of rH,
the PBL top was found at 2150 to 2200 m a.s.l.. Therefore the flight leg 7 at 2260 m a.s.l. and





leg 8 at 3020 m a.s.l. were performed in free tropospheric air. These two measurements show
a typical free tropospheric background concentration (~ 1.3 ng/m³, Weigelt et al., 2016 and
references therein). The measurements during the flight level change from leg 6 to leg 7
represent a mixture of boundary layer- and free tropospheric air (averaged altitude 2150 m
a.s.l.). Therefore the Tekran 1 GEM, Tekran 2 TGM, and Lumex GEM concentration of
2.3 ng m$^{-3}$, 2.4 ng m$^{-3}$, and 1.9 ng m$^{-3}$ was strongly influenced by the high concentration
below the boundary layer top.
In the altitude range 1600 m a.s.l. to 2200 m a.s.l. not only mercury, but also $SO_2$ was
significantly increased (from 1.6 ppb to 21.4 ppb), which clearly indicates that the mercury
was emitted from the CFPP. Inside the plume (leg 6), the $O_3$ concentration was slightly
decreased to 42.3 ppb. At the same time NO and $NO_2$ increased to 6.1 ppb and 8.9 ppb,
respectively. Outside the plume (e.g. leg 4) $O_3$ was 48.5 ppb, NO was below the detection
limit, and $NO_2$ was ~1.5 ppb. This indicates $O_3$ depletion due to NO oxidation taking place
inside the plume (cf. Fig. 3 and 4). The presence of a temperature inversion at the PBL top is
indicated by the changing T and θ vertical gradient in Fig. 5. This inversion layer prevents a
further ascent of the power plant plume. Therefore, the highest concentration of pollutants
was found below the PBL top. As already shown with Fig. 3 and 4, during a flight leg in a
certain altitude (and during level change) the aircraft did not remain within the plume all the
time. Therefore, the concentrations, given in Fig. 5 do represent a mixture of plume and
background air.
The ratio of concentration enhancements (ERs), $\Delta Hg/\Delta SO_2$, $\Delta Hg/\Delta CO$, and $\Delta NOx/\Delta SO_2$
represent the emission ratios at the stack if a) chemical reactions during the transport from the
stack to the point of interception can be neglected and b) the background concentrations have
not changed during the measurement including the transport from the stack to the place of
plume encounters. As mentioned above, the transport time from the stack to the location of
plume interception was ~ 0.9 h on both days. Based on OH concentrations measured in a
CFPP plume, Ambrose et al. (2015) estimated $SO_2$ and NOx lifetimes of 16 – 43 and 1.8 – 5.8
h, respectively. The combination of GEM, TGM, and GOM measurements by Lumex, Tekran
2537X (Tekran 1, with quartz wool trap), 2537B (Tekran 2, without quartz wool trap), and
KCl denuder, respectively, suggests that there is no substantial conversion of GEM into GOM
within the transport time of ~ 0.9 h. The vertical profile over Leipzig, upwind of the CFPP,
was measured on August 21 ~ 3 h after the measurements in the plume. The CO, $O_3$, $SO_2$,




NOx and Hg concentrations in the PBL over Leipzig with ~ 120, 50, 0.5, 3 ppb, 1.4 ng m$^{-3}$,
respectively, are similar to respective concentrations found outside of the plume over CFPP
Lippendorf. Differences between them for $SO_2$, NOx, and Hg are small when compared with
their enhancements in the plumes of ~ 40, 30 ppb, 4 ng m$^{-3}$, respectively. On August 22 no
vertical profile upwind was measured, but $SO_2$, NOx, and Hg concentrations over Waldhof, ~
90 km north of Leipzig, measured immediately before the downwind measurements of CFPP
Lippendorf, were comparable. We thus conclude that the background concentrations of $SO_2$,
NOx, and Hg have not changed significantly during the 0.9 h long transport from the stack to
the location of aircraft interception and during ~ 20 min of the repeated plume interceptions.
In addition, the large $SO_2$, NOx, and Hg enhancements in the plume make the calculated
$\Delta Hg/\Delta SO_2$ and $\Delta NOx/\Delta SO_2$ ERs insensitive to small changes in background $SO_2$, NOx, and
Hg concentrations. This is not always the case for small $\Delta CO$ and negative $\Delta O_3$ (negative
because $O_3$ is consumed by oxidation of NO to $NO_2$) relatively to their background mixing
ratios. In addition, the CO background mixing ratios changed substantially from ~123 to
105 ppb during the plume crossing #4 and #5 on August 21 due to altitude change. $\Delta Hg/\Delta CO$
for these plume interceptions was thus not calculated.
The ERs are usually calculated as a slope of Hg vs X correlations (e.g. Ambrose et al., 2015).
The advantage of this method is that the background concentrations of neither Hg nor X have
to be known as long as they remain constant during the measurement. The method, however,
is applicable only if the plume crossings are much longer than the response time of the
instruments. With the plume transects lasting in our case only 60 – 120 s and effective
temporal resolution of 10 s for $SO_2$ and NOx measurements, however, the signals have to be
carefully synchronized. In addition, the correlation slopes for individual plume crossings will
become quite uncertain because of small number of points. For this reason we apply the
correlation method for all (synchronized) points with $SO_2$ mixing ratios > 10 ppb. This
selection provides 35 and 45 points for Hg vs $SO_2$ correlations on August 21 and 22,
respectively. Individual plume crossings are not resolved by this calculation. Correlations
made by the bivariate Williamson-York method (Cantrell, 2008) provide a slope and its
statistical uncertainty representing ER (Hg/$SO_2$) and its uncertainty.
An alternative method calculates ERs as a ratio of $\Delta Hg$ to $\Delta X$ where $\Delta Hg$ and $\Delta X$ are signal
enhancements against the background integrated over the plume crossing. This method, called
here "integral method", is applicable for measurements with instruments with different





response times and we will show that it can use even Tekran measurements with a temporal
resolution of 150 s, although not for individual plume crossings. Opposite to the correlation
method, no exact synchronization is needed. The disadvantage, however, is that the results are
sensitive to the selection of background concentrations. Figures 3 and 4 show that background
Hg concentrations are especially difficult to define from the Lumex measurements. We thus
use the Hg background concentrations measured by the much more precise Tekran instrument.
As the Lumex instrument measured only GEM, we use the background measured by Tekran
instrument with quartz wool (Tekran 1). The other disadvantage of the integral method is that,
opposite to the correlation method, the uncertainty of ERs is difficult to quantify. We
overcome this difficulty here by averaging the ERs from individual plume crossings and
taking the standard deviation as a measure of ER uncertainty.
The $Hg/SO_2$ ERs are listed in Table 2. The correlation and integral methods provide similar
results with $5.53 \pm 1.10$ and $5.56 \pm 1.19$ µmol mol$^{-1}$, respectively, for August 21, and $7.38 \pm$
$0.92$ and $6.32 \pm 1.52$ µmol mol$^{-1}$, respectively for August 22. The integral method with Tekran
and $SO_2$ integrals over all plume encounters provide somewhat higher $Hg/SO_2$ ERs but still
within the uncertainties of the correlation and integral methods. The measured $Hg/SO_2$ ERs
are smaller than the emission ratio of 10.8 µmol mol$^{-1}$ calculated from Hg and $SO_2$ annual
emissions reported by the operator for 2013. They are close to $5.2 – 6.5$ µmol mol$^{-1}$
determined by Ambrose et al. (2015) for Big Brown (BBS) and Dolet Hills Stations (DHS).
BBS, a 1187 MW CFPP in Texas, is fired with subbituminous coal and is equipped with
activated carbon injection flue cleaning. DHS, a 721 MW CFPP in Louisiana, is fired with
lignite and is equipped with wet flue gas desulfurization, similar to CFPP Lippendorf.
$Hg/CO$ ERs are frequently used to classify the origin of different plumes (Slemr et al., 2009,
2014; Lai et al., 2011) with ERs < 0.25 µmol mol$^{-1}$ typical for plumes from biomass burning
and ERs > 0.6 µmol mol$^{-1}$ characteristic for plumes of urban/industrial origin. The $Hg/CO$
ERs measured in the plume of CFPP Lippendorf are listed in Table 3. The correlation method
tends to yield somewhat higher $Hg/CO$ ERs than the integral method. Because of changing
background on August 21 and changing altitude on August 22, no ERs were calculated by
integral method using the Tekran measurements. As mentioned before, the high background
CO mixing ratios and relatively small CO enhancement in the plume make the integral
method quite sensitive to the chosen background. For this reason we believe 5.2 and 9.4 µmol
mol$^{-1}$ from correlation method for August 21 and August 22, respectively, to be more reliable.





The Hg/CO emission ratio from the 2013 annual emissions reported by the operator is 7.6
μmol mol$^{-1}$, in reasonable agreement with our measurements. Hg/CO ERs of this magnitude
have never been observed so far in the plumes detected during the CARIBIC flights (Slemr et
al., 2014). This is probably because only large plumes extending over several hundreds to few
thousands of km can be detected by these flights. Their Hg/CO ERs are then a mixture of
Hg/CO ERs from point sources embedded in plumes from larger industrial and/or urban areas.
Simultaneous NOx and SO$_2$ measurements allow us to calculate also the NOx/SO$_2$ ERs which
are listed in Table 4. The ERs from the correlations and integral methods are in good
agreement with each other on both days. The NOx/SO$_2$ ER of 0.59 mol mol$^{-1}$ on August 21 is
almost twice as large as 0.27 mol mol$^{-1}$ on August 22, and both ERs are substantially lower
than the emission ratio of 0.91 mol mol$^{-1}$ calculated from the NOx and SO$_2$ emissions reported
for 2013. All these NOx/SO$_2$ ERs are substantially larger than ~0.08 mol mol$^{-1}$ reported by
Ambrose et al. (2015) for Big Brown CFPP in Texas and corrected for the NOx loss during
the transport from the stack to the point of the plume interception.
Ozone is not emitted but the ambient O$_3$ is consumed by a rapid reaction with NO (O$_3$ + NO =
NO$_2$ + O$_2$) in the plume during the transport from the stack to the point of plume interception.
The O$_3$/NOx ERs thus do not represent emission ratios and they are negative because of O$_3$
consumption. If only NO were emitted the O$_3$/NOx ER should be -1 mol mol$^{-1}$. O$_3$/NOx ERs
were not calculated for August 21 because of changing O$_3$ background mixing ratio. The
calculated O$_3$/NOx ERs for August 22 are listed in Table 5. The correlation method provides a
slope of -0.62 ± 0.13 mol mol$^{-1}$ while the integral method provides an ER of -1.0 ± 0.6 mol
mol$^{-1}$. We thus conclude that the emitted NO constitute some 60 – 100% of NOx emissions.
**4 GOM emissions**
As mentioned earlier the GOM measurements made here using quartz wool traps and KCl
denuders can be both influenced by high humidity (Huang and Gustin, 2015) and those made
by KCl additionally by high O$_3$ concentrations (Lyman et al., 2010). Because of NO
emissions, the O$_3$ concentrations in the CFPP plumes will be lower than in ambient air making
this interference unlikely. The humidity interference would lead to an underestimation of
GOM concentrations measured by KCl denuders and overestimation of GEM concentrations
measured by Tekran instrument with quartz wool trap. However, specific GEM measurements



are provided by Lumex, an atomic absorption instrument with Zeeman background correction,
albeit with a worse precision when compared to Tekran measurements.
Table 6 lists the GOM concentrations measured by the KCl denuders during the vertical
profiles over Leipzig and in the plume of CFPP Lippendorf on August 21, 2013, and over
Waldhof on August 22, 2013. Taking into account the uncertainty of $\pm$ 5 pg m$^{-3}$ there is hardly
any difference between GOM concentration of 5.8 pg m$^{-3}$ measured during the vertical profile
over Leipzig and 11.4 pg m$^{-3}$ in the plume of CFPP Lippendorf on August 21. The difference
of 5.6 pg m$^{-3}$ is distributed over the vertical profile of 3000 m. Assuming ~ 300 m thick layer
with the CFPP plume and nearly zero GOM concentrations outside of this layer, the GOM
concentrations in the layer would be ~ 60 pg m$^{-3}$. This is roughly consistent with the
differences between Tekran measurements without quartz wool trap and with it. The average
difference in the plume was 87 $\pm$ 117 pg m$^{-3}$ (n=8) on August 21 and 63 $\pm$ 79 pg m$^{-3}$ (n=12).
Related to the average TGM enhancement (Tekran without quartz wool trap) in the plume of
0.90 ng m$^{-3}$ on August 21 and of 1.03 ng m$^{-3}$ on August 22, the GOM concentration would
represent ~ 10% and ~ 6% of TGM emissions on August 21 and 22, respectively.
An independent assessment of the GOM emissions can be made using Hg/SO$_2$ ERs listed in
Table 2. On August 21, the Hg/SO$_2$ ER of 5.5 $\pm$ 1.1 µmol mol$^{-1}$ from correlation and 5.6 $\pm$ 1.2
µmol mol$^{-1}$ from integral methods, both based on specific GEM measurements by Lumex, are
within their uncertainties consistent with 6.6 µmol mol$^{-1}$ derived from Tekran with quartz
wool trap. On August 22, the Hg/SO$_2$ ER of 7.4 $\pm$ 0.9 µmol mol$^{-1}$ from correlation method is
consistent with 8.1 µmol mol$^{-1}$ determined from Tekran data, while the 6.3 $\pm$ 1.5 µmol mol$^{-1}$
from the integral method is somewhat lower. Consequently, Hg/SO$_2$ ERs from less specific
measurements with quartz wool trap tend to be somewhat higher but within their combined
uncertainties comparable with those derived from GEM specific Lumex measurements. A
comparison of Hg/SO$_2$ ERs measured by Tekran without and with quartz wool trap implies
GOM emissions representing 13 and 9% of TGM emissions on August 21 and 22,
respectively. Taking GEM specific Lumex measurements instead of those made by Tekran
with quartz wool trap would imply GOM emissions representing 27 and 24% on August 21
and 22, respectively, which we consider an upper limit.
In summary we conclude that GOM represents ~ 10% of the TGM emitted from CFPP
Lippendorf with an uncertainty range of 0 - 25%. Edgerton et al. (2006) reported GOM
fraction of 13, 19, and 21% of total mercury in the plumes from CFPPs Hammond, Crist, and



Bowen in the U.S. Stergašek et al. (2008) reported 4% GOM fraction for Hg emissions from
CFPP with FGD in Slovenia which was fired by lignite. Wang et al. (2010) found GOM
fractions of 6 -25% of all Hg emissions from five Chinese power plants with FGD. Deeds et
al. (2013) found 13% of total mercury being GOM in the plume of CFPP Nanticoke in
Canada. They think that discrepancy between this and 43% GOM fraction found in stack
gases is due to sampling biases. Tatum Ernest et al. (2014) support their findings using a
speciation technique still in development. On the other side Landis et al. (2014) report high
GOM fractions of > 86% in stack gases of the Crist CFPP and 4 – 40% conversion into GEM
in the plume in 0.6 – 1.3 km distance from the stack. They attribute the difference to a
reduction of GOM to GEM. Putting this unresolved issue aside, low fractions of GOM
emissions reported here and by others (Edgerton et al., 2006; Stergašek et al., 2008; Wang et
al., 2010; Deeds et al., 2013; Landis et al., 2014) are in contrast to the AMAP/UNEP
geospatially distributed mercury emissions dataset "2010v1" (Wilson et al., 2013), splitting
the speciated mercury emissions from combustion in power plants to 50% GEM, 40% GOM,
and 10% PBM. As mentioned before the flue gas desulphurisation (FGD) in CFPP Lippendorf
is made by washing of the flue gas with CaO suspension and this type of FGD is known to
capture most of GOM (Schütze, 2013). Although no PBM was measured in this study, 10% of
mercury being emitted as PBM according to the inventory is also an overestimation for CFPPs
with FGD (Stergašek et al., 2008; Wang et al., 2010).

## 5 Conclusions

Plume of the coal fired power plant (CFPP) Lippendorf near Leipzig in Germany was
encountered several times on August 21 and 22, 2013. On August 21 the plume was captured
at below planetary boundary layer top due to a temperature inversion layer. $Hg/SO_2$, $Hg/CO$,
$NOx/SO_2$ ERs in the plume were determined as a slope of bivariate correlations of the species
concentrations and as ratios of integrals over the individual plume crossings.  The measured
$Hg/SO_2$ and $Hg/CO$ ERs were, within the measurement uncertainties, consistent with the ERs
calculated from annual emissions reported by the CFPP operator for 2013, the $NOx/SO_2$ ER
was somewhat lower.
GOM fraction of total mercury emissions was estimated a) using GOM measurements by KCl
denuders, b) from a difference between Hg measurements by Tekran instruments without and
with quartz wool trap, and c) from a difference between Hg measurements by a Tekran



instrument without quartz wool trap and GEM specific measurements by Lumex instrument.
Despite large uncertainties in all these estimates we conclude that GOM emissions represent
~10% of total mercury emissions with an uncertainty range of 0 – 25%. This result is
consistent with findings by others (Edgerton et al., 2006; Stergašek et al., 2008; Wang et al,
2010; Deeds et al., 2013) and suggests that GOM fractions of ~40% of CFPP mercury
emissions in current emission inventories are overestimated. Although PBM was not
measured by us, its inventoried fraction of 10% is too high too for CFPPs with FGD
according to the above references.

## Acknowledgements

Measurements were carried out as part of the European Tropospheric Mercury Experiment
(ETMEP) within the Global Mercury Observation System project (GMOS; www.gmos.eu).
GMOS is financially supported by the European Union within the seventh framework
programme (FP-7, Project ENV.2010.4.1.3-2). Special thanks to Compagnia Generale
Ripreseaeree (http://www.terraitaly.it/) in Parma/Italy and the pilots Oscar Gaibazzi and Dario
Sassi for carrying out the measurement flights.





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



**Tables**
Table 1: List of instruments, installed into the CASA 212 research aircraft. The acronyms are:
GEM = gaseous elemental mercury; GOM = gaseous oxidized mercury.

| Parameter | Instrument name | Temporal resolution | Uncertainty | Lower detection limit |
|---|---|---|---|---|
| GEM | Lumex RA-915AM (modified, T-stabilised by Lumex company) | 1 sec (raw signal) | ±4 ng/m³ (1 s raw signal) ±1 ng/m³ (10 s average) | 0.5 ng/m³ (120 s average) |
| GEM | Tekran: 2537X (with upstream quartz wool trap) | 150 s | ±12.5% of reading | 0.1 ng·m$^{-3}$ |
| GEM + unknown amount of GOM* | Tekran 2537B | 150 s | ±12.5% of reading | 0.1 ng·m$^{-3}$ |
| GOM | manually denuder samples | 2600 to 3600 s | ±5 pg·m$^{-3}$** | 1 pg·m$^{-3}$ |
| CO | Aero Laser AL5002 | 1 s | ±3% of reading | 1.5 ppb |
| $O_3$ | Teledyne API 400E | 10 s | ±2% of reading | 0.6 ppb |
| $SO_2$ | Thermo: 43C Trace Level | 10 s | ±4% of reading | 0.2 ppb |
| NO NO$_2$ | Teledyne API M200AU | 10 s 10 s | ±10% of reading | 0.05 ppb |
| Pressure | Sensor Technics CTE7001 | 1 s | ±1% of reading | 0 mbar |
| Temperature | LKM Electronic DTM5080 | 1 s | ±0.13°C | -50°C |
| Relative Humidity (rH) | Vaisala HMT333 | 8 s | ±1.0% rH (0-90% rH) ±1.7% rH (90-100% rH) | 0% |
| GPS data (3d position, speed, heading) | POS AV | 1 s | ±5 m (horizontal)*** ±15 (vertical)*** | --- |

* The aircraft inlet system transmission efficiency for GOM was not tested because no GOM sources
were available which would enable measurements during the flight.
** Difference of the two blank tests
*** The GPS accuracy is dependent on the number of satellites. The given numbers are estimated
values.





Table 2: Hg/SO$_2$ enhancement ratios (ERs). Correlation method: 10 s average Hg
concentrations measured by Lumex correlated with 10 s average SO$_2$ mixing ratios, only Hg
values with SO$_2$ concentrations > 10 ppb were taken, uncertainties set to 1 ng m$^{-3}$ for Lumex
and 0.5 ppb for SO$_2$. Integral method: 1 s Lumex and SO$_2$ signals integrated over the duration
of Lumex measurement, ̶measurements of Tekran with quartz wool taken as Lumex
background concentrations (i.e. 1.27 and 1.25 ng m$^{-3}$ for August 21 and 22, respectively). SO$_2$
background mixing ratio was 0.83 and 0.66 ppb on August 21 and 22, respectively.

| Date | Method | Species | ER 10$^{-6}$ mol mol$^{-1}$ | n, R, signif | Comment |
|---|---|---|---|---|---|
| August 21, 2013 | correlation | GEM | 5.53 ± 1.10 | 35, 0.6564, >99.9% | |
| | integral peak 1 | GEM | 6.67 | | Lumex zeroing |
| | integral peak 2 | GEM | 5.72 | | |
| | integral peak 3 | GEM | 5.98 | | Lumex zeroing |
| | integral peak 4 | GEM | 3.88 | | |
| | integral peak 5 | GEM | 0.89 | | |
| | integral average | GEM | 5.56 ± 1.19[*] | 4[*] | |
| | Tekran with quartz wool trap | GEM | 6.56 | | |
| | Tekran | TGM | 7.55 | | |
| August 22, 2013 | correlation | GEM | 7.38 ± 0.92 | 45, 0.7751, >99.9% | |
| | integral peak 1 | GEM | 6.44 | | |
| | integral peak 2 | GEM | 4.83 | | |
| | integral peak 3 | GEM | 5.90 | | Lumex zeroing |
| | integral peak 4 | GEM | 6.67 | | |
| | integral peak 5 | GEM | 9.03 | | Lumex zeroing |
| | integral peak 6 | GEM | 5.02 | | |
| | integral average | GEM | 6.32 ± 1.52 | 6 | |
| | Tekran with quartz wool trap | GEM | 8.13 | | |
| | Tekran | TGM | 8.97 | | |
| 2013 | reported annual emissions | TGM | 10.8 | | |

[*]average without integral of peak 5 which is identified as outlier by Nalimov test (at >95%
significance level, Kaiser and Gottschalk, 1972)





Table 3: Hg/CO enhancement ratios (ERs). Correlation method: 10 s average Hg
concentrations measured by Lumex correlated with 10 s average CO mixing ratios for $SO_2$
mixing ratios above 10 ppb, uncertainties set to 1 ng m$^{-3}$ for Lumex and 1 ppb for CO.
Integral method: 1 s Lumex and CO signals integrated over the duration of Lumex
measurement, Tekran 1 readings taken as Lumex background concentrations (i.e. 1.27 and
1.25 ng m$^{-3}$ for August 21 and 22, respectively). CO background mixing ratio was 119.3 ppb
on August 21 and 123.8 ppb on August 22.

| Date | Method | ER (Hg/CO) | | Comment |
|---|---|---|---|---|
| | | $10^{-5}$ mol mol$^{-1}$ | n, R, signif | |
| August 21, 2013 | Correlation | 5.19 ± 0.94 | 31, 0.6596, >99.9% | values only until 10:40:20 |
| | integral peak 1 | 3.40 | | Lumex zeroing |
| | integral peak 2 | 4.16 | | |
| | integral peak 3 | 3.33 | | Lumex zeroing |
| | integral peak 4 | | | background change |
| | integral peak 5 | | | CO calibration |
| | integral average | 3.63 ± 0.46 | 3 | |
| August 22, 2013 | Correlation | 9.43 ± 1.07 | 37, 0.7880, >99.9% | |
| | integral peak 1 | 3.19 | | |
| | integral peak 2 | | | CO calibration |
| | integral peak 3 | | | Lumex zeroing, CO calibration |
| | integral peak 4 | 7.87 | | |
| | integral peak 5 | 5.61 | | Lumex zeroing |
| | integral peak 6 | 4.75 | | |
| | integral average | 5.36 ± 1.95 | 4 | |
| 2013 | reported annual emissions | 7.58 | | |



Table 4: NOx/SO$_2$ enhancement ratios (ERs). Correlation method: 10 s average NOx mixing
ratios correlated with 10 s average SO$_2$ mixing ratios above 10 ppb, uncertainties set to 1 ppb
for NOx and 0.5 ppb for SO$_2$. Integral method: 1 s NOx and 1 s SO$_2$ signals integrated over
the duration of the individual plume intersection, background mixing ratios for SO$_2$ and NOx
are 0.83 and 1.78 ppb, respectively, for August 21 and 0.66 and 0.45 ppb, respectively for
August 22.

| Date | Method | ER (NOx/SO$_2$) mol mol$^{-1}$ | n, R, signif | Comment |
|---|---|---|---|---|
| August 21, 2013 | Correlation | 0.585 ± 0.038 | 34, 0.9379, >99.9% | |
| | integral peak 1 | 0.598 | | |
| | integral peak 2 | 0.575 | | |
| | integral peak 3 | 0.725 | | |
| | integral peak 4 | 0.497 | | |
| | integral peak 5 | | | |
| | integral average | 0.598 ± 0.095 | 4 | |
| August 22, 2013 | Correlation | 0.262 ± 0.051 | 40, 0.6344, >99.9% | |
| | integral peak 1 | 0.297 | | |
| | integral peak 2 | 0.457 | | |
| | integral peak 3 | 0.167 | | Lumex zeroing |
| | integral peak 4 | 0.330 | | |
| | integral peak 5 | 0.133 | | Lumex zeroing |
| | integral peak 6 | 0.317 | | |
| | integral average | 0.284 ± 0.118 | 6 | |
| 2013 | reported annual emissions | 0.910 | | |





Table 5: $O_3$/NOx enhancement ratios (ERs). Correlation method: 10 s average $O_3$ mixing
ratios correlated with 10 s average $SO_2$ mixing ratios above 10 ppb, uncertainties set to 1 ppb
for $O_3$ and 1 ppb for NOx. Integral method: 1 s $O_3$ and 1 s NOx signals integrated over the
duration of the individual plume intersection, background mixing ratios for $O_3$ and NOx are
43.09 and 1.78 ppb, respectively, for August 21. Individual $O_3$ background mixing ratios
(average of background before and after the peak) varying between 53.9 ppb for peak 1 to
56.2 ppb for peak 4 were taken for August 22. The NOx background mixing ratio on August
22 was 0.45 ppb.

| Date | Method | ER ($O_3$/NOx) | | Comment |
|---|---|---|---|---|
| | | mol mol$^{-1}$ | n, R, signif | |
| August 22, 2013 | Correlation | -0.620 ± 0.134 | 40, -0.3776, >95% | |
| | integral peak 1 | -0.979 | | |
| | integral peak 2 | -0.424 | | |
| | integral peak 3 | -1.527 | | |
| | integral peak 4 | -0.686 | | |
| | integral peak 5 | -2.059 | | |
| | integral peak 6 | -0.568 | | |
| | integral average | -1.040 ± 0.633 | 6 | |



Table 6: Results of the manual KCl denuder samples during all ETMEP-2 measurement
flights in 2013 over central Europe. GOM data were corrected for denuder blank test,
additionally performed over Iskraba/Slovenia and Waldhof/Germany. GOM concentrations
are given as a centre of an estimated uncertainty range (in brackets) and are given at standard
temperature and pressure (STP; T=273.15 K, p=1013.25 hPa).

| Date | Location | Profile character (relative sampling time in PBL* and FT** air | analysed GOM concentration [pg m$^{-3}$] |
|---|---|---|---|
| 2013-08-21 | Lippendorf/Germany | vertical (76% PBL: 24% FT) | 11.4 (7.0-15.7) |
| 2013-08-21 | Leipzig/Germany | vertical (61% PBL; 39% FT) | 5.8 (1.0*** - 10.6) |
| 2013-08-22 | Waldhof/Germany | vertical (54% PBL; 46% FT) | 31.0  (24.6-37.3) |

* planetary boundary layer (PBL)
** free troposphere (FT)
***If a concentration was found to be below the method lower detection limit of 1.0 pg m$^{-3}$,
the lower detection limit is given.



**Figures**

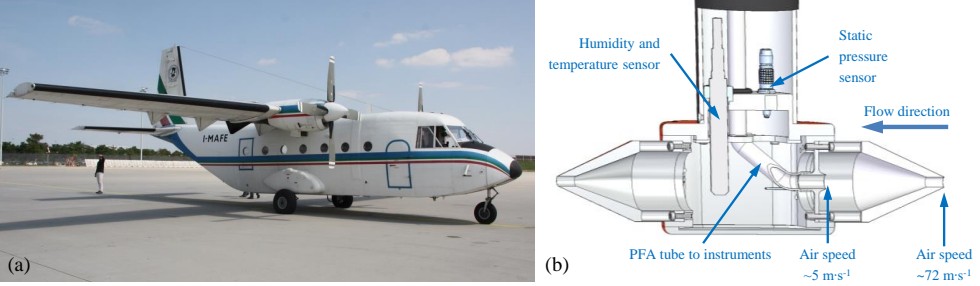

(a)                                                                (b)
Figure 1: For the ETMEP-2 campaign in August 2013 the CASA 212 (a) from the Italian
company Compagnia Generale Ripreseaeree (http://www.terraitaly.it/) was equipped with
specially designed and manufactured trace gas inlet (b).



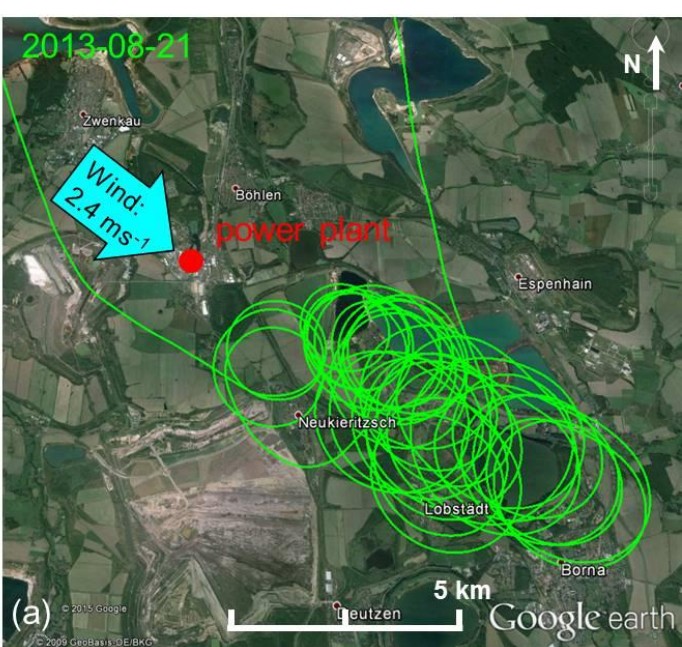

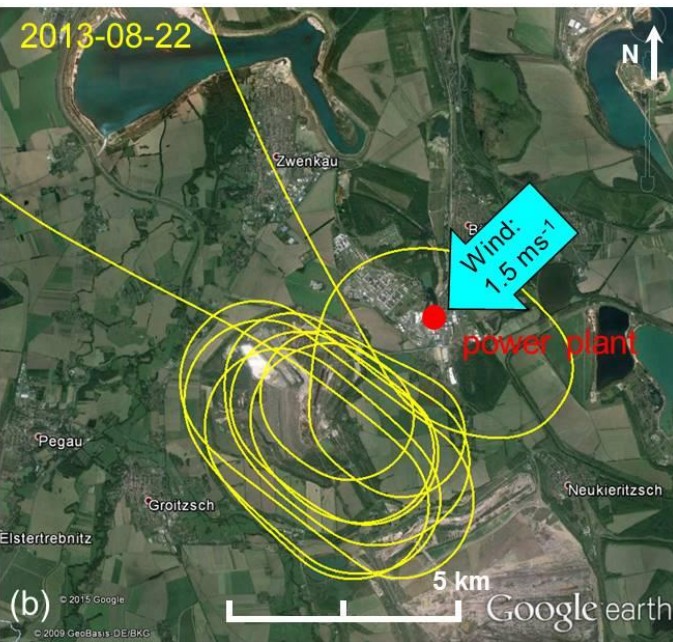

3    Figure 2: Flight track of the ETMEP-2 flights on August 21 (a) and 22 (b), 2013 downwind

4    the coal fired power plant "Lippendorf", south of Leipzig, Germany. On both flights the

5    power plant plume was crossed several times.



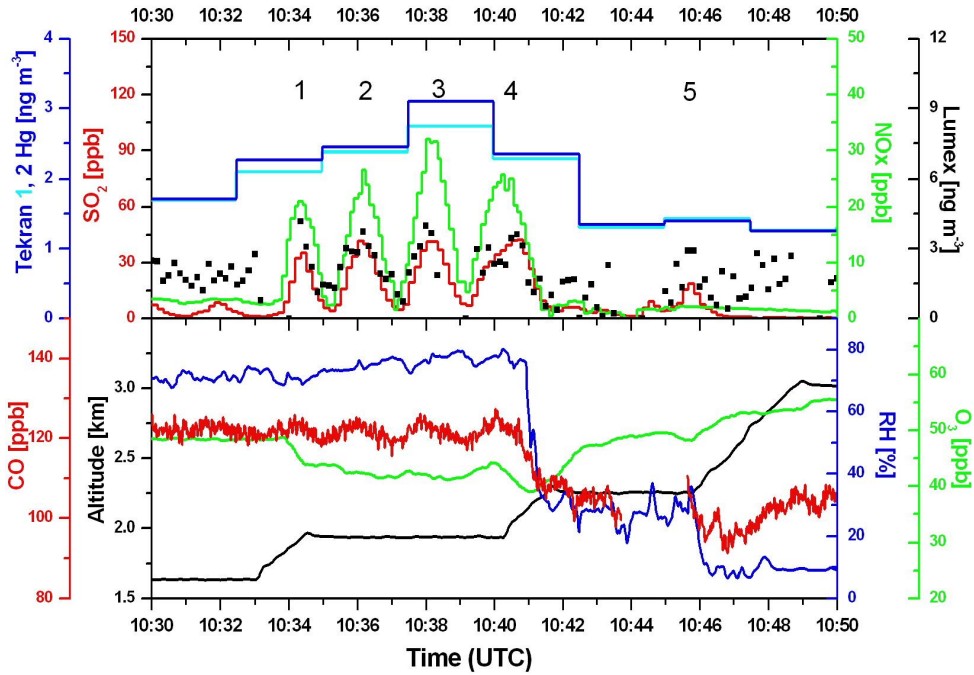

Figure 3: ETMEP-2 coal fired power plant plume measurements on August 21, 2013 south of
Leipzig/Germany.  The gaps in the Lumex signal (10 s resolution) are due to internal zero air
checks for the correction of the instruments base line drift. Tekran 1 was run with quartz wool
trap at the inlet of the instrument presumed to remove GOM, Tekran 2 without. Tekran 1 and
2 measurements are thus presumed to represent GEM and TGM measurements, respectively.
All parameters were synchronized using individual instrument delay and response times. All
Hg concentrations are given at standard temperature and pressure (STP; T=273.15 K,
p=1013.25 hPa).





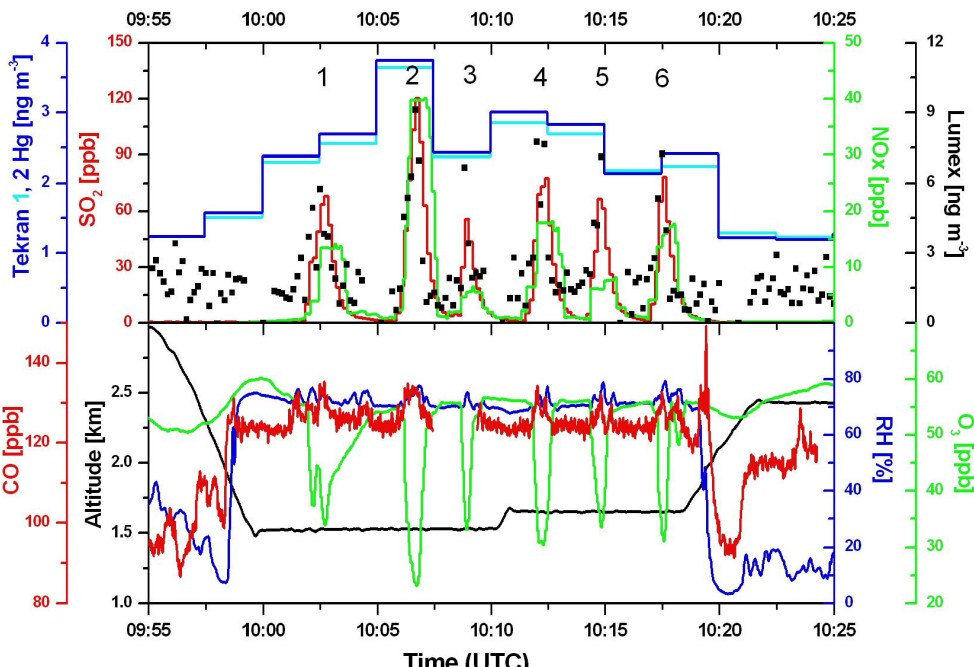

Figure 4: ETMEP-2 coal fired power plant plume measurements on August 22, 2013 south of
Leipzig/Germany. The gaps in the Lumex signal (10 s resolution) are due to internal zero air
checks for the correction of the instruments base line drift. Tekran 1 was run with quartz wool
trap at the inlet of the instrument presumed to remove GOM, Tekran 2 without. Tekran 1 and
2 measurements are thus presumed to represent GEM and TGM measurements, respectively.
All parameters were synchronized using individual instrument delay and response times. All
Hg concentrations are given at standard temperature and pressure (STP; T=273.15 K,
p=1013.25 hPa).





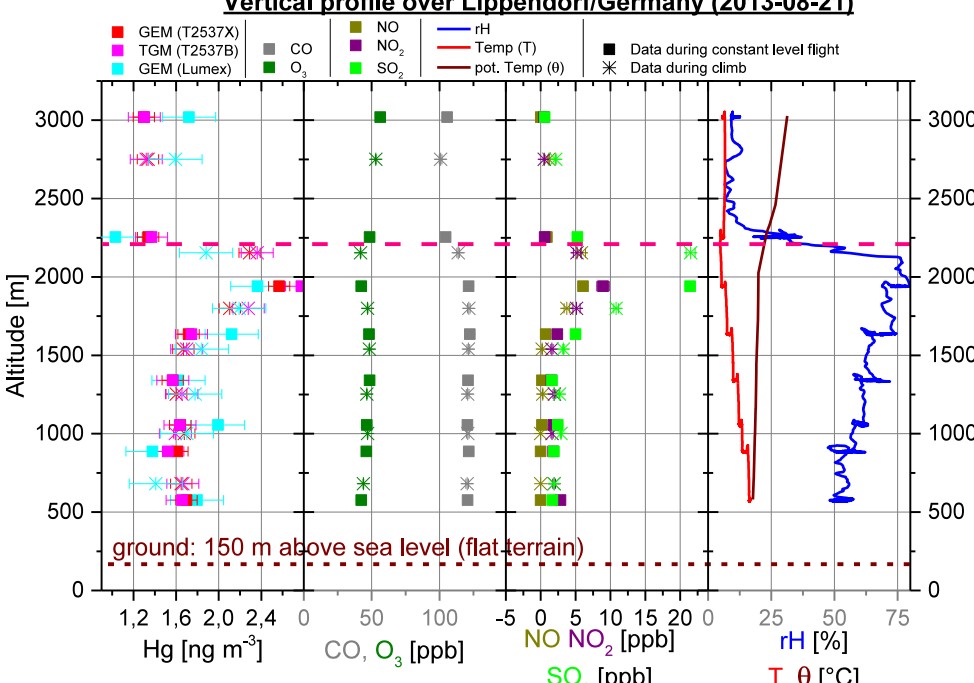

Figure 5: Vertical profile, measured on 21 August 2013 from 13:17:30 to 14:07:30 (local

time) downwind the coal fired power plant Lippendorf (central Germany; 45.561°N,

14.858 °E, elevation: 150 m a.s.l.; flat terrain). Squares represent 300 s averages with

horizontal flight leg; stars indicate 150 s averages during climbing between two neighbouring

flight legs. The red dashed line indicates the planetary boundary layer (PBL) top, which was

determined to be at 2150 to 2250m a.s.l.. All Hg concentrations are given at standard

temperature and pressure (STP; T=273.15 K, p=1013.25 hPa).