# Peer review of "Andreas Weigelt1,\*, Franz Slemr2, Ralf Ebinghaus1, Nicola Pirrone3, Johannes"

_Atmospheric Chemistry and Physics, 2016_

## Referee Comment (RC1) · Anonymous Referee #1 · 20 Jul 2016

Summary:

This manuscript reports measurements of gaseous elemental mercury (GEM), total gaseous mercury (TGM), gaseous oxidized mercury (GOM), carbon monoxide (CO), nitrogen oxides (NOx), sulfur dioxide (SO2), ozone (O3), and meteorological parameters (temperature, pressure, relative humidity, and wind velocity) in the vicinity of Leipzig, Waldhof, and the coal-fired power plant (CFPP) Lippendorf in Germany during August 21 and 22, 2013. The measurements were made from on board a research aircraft and were used to derive Hg/CO, Hg/SO2, and NOx/SO2 emission ratios (ERs) inside the Lippendorf plume. The measurement-based Hg/CO, Hg/SO2, and NOx/SO2 ERs are compared with values calculated from emissions data reported for the Lippendorf CFPP. The Hg measurements and the Hg/SO2 ERs are further used to estimate GOM/TGM ERs for the Lippendorf CFPP. The main conclusions of the study are that

measured Hg/CO and Hg/SO2 ERs are consistent with values reported by the CFPP, whereas observed GOM/TGM ERs are lower than a contemporary global inventory reports. The latter finding is mostly consistent with results of recent field studies. However, one recent study (Landis et al., 2014) reported much higher GOM fractions in a CFPP plume and showed evidence for GOM reduction during downwind plume transport.

Top down verification of Hg emissions from CFPPs is an important research goal. The results and analysis presented by the authors are valuable to better understanding anthropogenic Hg emissions. My main criticisms (detailed under "Specific Comments") are that too little experimental detail is provided on the Hg measurements, in particular the sampling and analysis of GOM. Also, I question the validity/significance of the estimated NO2/NOx ERs. I also feel that further discussion/analysis is needed to relate the authors' GOM/TGM ERs to other values reported in the literature and to the AMAP/UNEP Hg emissions inventory. Additionally, I suggest that the authors seek assistance to correct numerous grammatical errors in the paper. (Some, but not all, grammatical errors are identified below.)

I would recommend that this manuscript be published in ACP after the authors address my "Specific Comments".

Specific Comments:

Page 2 1. Line 1: It is preferable to define compounds when they are introduced (e.g., sulfur dioxide (SO2), carbon monoxide, (CO), etc.). 2. Line 4: (Grammatical error) Insert "The" at the beginning of the sentence starting with "GOM". 3. Line 4: "GOM" should be defined when first used. In general, the authors should check that all acronyms are defined when introduced. 4. Lines 5–6: (Grammatical error) Insert commas (,) after "were" and "uncertainties". 5. Lines 5–7: (Grammatical error) The sentence beginning with "Measured" is a run-on. This could be corrected by inserting "while" or "whereas" after the last comma. 6. Line 9: Please add some information to

qualify the "~10%" value. For example, is this an average or median value? How many individual GOM/TGM ER measurements are represented? If this is an average, the standard deviation, confidence interval, or standard error should be included. 7. Line 16: I suggest you replace "its" with "Hg". 8. Line 24: Insert "Hg" before "emissions". 9. Line 28: Recent work suggests the atmospheric lifetime of elemental Hg may be considerably shorter than 1 yr, though Hg may cycle between elemental and oxidized forms multiple times before being (wet or dry) deposited. I suggest citing a range of lifetimes and adding some more relevant references from the recent literature (e.g., Shah et al., 2016, http://www.atmos-chem-phys.net/16/1511/2016/). 10. Line 29: I suggest rewording the end of the sentence to replace "PM" with "PBM". The acronym "PM" is typically used for "particulate matter". I suggest using "PBM", which is more common in the Hg literature, throughout the manuscript. 11. Line 29: In addition to "washed out" and "rained out" I would include "dry deposited", as this is also an important pathway for GOM (and PBM) removal from the atmosphere.

Page 3 12. Line 9: For clarity, I suggest you insert "in-plume" before "GEM oxidation". 13. Lines 12–14: It is not generally true that power plant operators are required to measure and report their emissions. Regulations requiring monitoring and reporting of Hg emissions from CFPPs only recently were enacted in the U.S. I suggest you state that direct measurements of total and speciated Hg emissions from CFPPs are somewhat limited. Then you could state that our understanding of Hg emissions from CFPPs is further limited by the complicating factors you identify (e.g., variable coal composition, complex flue chemistry, etc.). 14. Line 14: "burnt coal" should be "coal burned".

Page 4 15. Lines 6–8: I suggest you add a reference in support of the claim that Lippendorf is "one of the most modern and efficient CFPPs in Europe". 16. Lines 9–13: I suggest you specify which pollutants were used to classify Lippendorf as the "4th most harmful" and "14th most harmful" emitter. 17. Lines 16–19: Please add some more information to define "LEV". I'm assuming the LEV refers to an in-stack, posttreatment concentration, but it would help if this were clearly stated. 18. Lines 19–20: Emissions monitoring "is mandatory" in Germany or throughout the EU? Many readers will not be familiar with relevant Hg regulations in the study region. Please be more specific. 19. Line 27: Insert a period (.) at the end of the sentence. 20. Lines 30–31: It seems that once you have defined "normal cruising speed" (i.e., line 25), you don't need to restate the value.

Page 5 21. Line 15: Are you referring to the tubing I.D. or O.D.? 22. Line 25: See comment 20.

Page 6 23. Lines 20–31: Too little information is given about how the denuder sampling and analysis was carried out. What was the inlet system that was used for the denuders? The authors should add information to describe the laboratory denuder analysis. How exactly was the analysis performed? Were the denuders loaded in a Tekran 2537/1130 system or was another desorption/analyzer configuration used? Diagrams of both the aircraft sampling system and the laboratory analysis system would be helpful.

Page 7: 24. Line 8: "nitric dioxide" should be "nitrogen dioxide". These compounds should be defined when introduced (see comment 3). 25. Lines 20–29: Too little information is given on calibration and zeroing of the Hg instruments. Were the internal permeation sources verified with a primary source (e.g., a Tekran 2505)? How were the instruments/inlet system zeroed?

Page 8 26. Lines 10–17: It would help to include a map showing the locations of Lippendorf, Leipzig, and Waldhof together. You could include Hamburg or refer to its direction on the map. It seems it would be helpful to show (or point in the direction of) Leipzig in Fig. 2. 27. Line 11: Remove "a" before "CFPP". 28. Line 27: NOx should have already been defined earlier.

Page 9 29. Line 9: Insert "of" after "downwind". 30. Lines 27–31: I don't think the discussion is strengthened by discussing the "flight level change" measurements here.

The "horizontal flight legs" data seem sufficient to demonstrate the vertical gradient above the 5th flight leg in the PBL.

Page 10 31. Lines 8–10: The fact that the wind direction also points to Lippendorf is also significant.

Page 12 32. Lines 14–16: It is not clear from the text or Table 2 exactly how the Hg/SOÂň2 ERs were calculated from the Tekran Hg measurements. It seems that $\triangle$Hg/$\triangle$SO2 was calculated first for each plume encounter and then all values were averaged. If this was the case, why are the number of measurements and the standard deviation (or standard error) not given in Table 2? Some additional details need to be added to Table 2 to explain that the Tekran-based ERs were calculated using the integral method. 33. Line 22: How does the estimated 0.6 to 1.0 NO2/NOx ratio compare to other estimates/observations for similar power plants (c.f., Peischl et al., 2010, http://onlinelibrary.wiley.com/doi/10.1029/2009JD013527/abstract )? A NO/NOx emission ratio of 0.6 would be quite low. You should explain additional assumptions that factor into your estimate. For instance, the observed O3/NOx ratio would seem to be sensitive to the fate of NO2. I question whether your calculation is completely valid or useful.

Page 14 34. Line 8: How did you estimate the thickness of the plume? This should be explained. If you can provide some uncertainty in this number you should be able to estimate uncertainty in the calculated GOM/TGM values, assuming GOM concentrations were in fact negligible outside of the plume. 35. Lines 24–29: Estimated uncertainties should be included with the GOM/TGM percentage values quoted.

Pages 14–15 36. Page 14, Lines 31–32; Page 15, Lines 1–19: It would be helpful to remind the reader at what distances/transport times from the Lippendorf CFPP you encountered the plume. How do these distances/transport times compare with those of the past studies cited? In particular, could differences in sampling (i.e., transport time) reconcile the authors' results with the findings of Landis et al. (2014)? Might

the inventory-based GOM fraction still be relevant, but only for very fresh plumes? How representative of the mix of CFPPs represented in the in Wilson et al. (2013) inventory is the Lippendorf CFPP? In other words, might we expect Lippendorf to be a relatively low GOM emitter? A more detailed discussion here would add significant value to the results. Without further consideration of whether the authors' results could be consistent with those of Landis et al. (2014), or whether Lippendorf is expected to be a relatively low GOM emitter compared to the Wilson et al. (2013) inventory, the conclusions that GOM/TGM is overestimated in the inventories may be misleading. 37. Lines 3–8: See comment 36 (also relevant to the abstract, lines 9–11).

Pages 24 and 29 38. Tables 1 and 6: It isn't clear to me why is the GOM "method lower detection limit" is so much lower than the uncertainty. I suggest you explain how the "method lower detection limit" for GOM was estimated.

Page 32 39. Lines 6–8: I suggest you change "Tekran 1, 2 Hg" in the figure to "GEM, TGM" and eliminate from the caption the explanation of how each Tekran was configured. This explanation was already provided in the text.

Page 33 40. Lines 6–8: See comment 38.

Page 34 41. Fig. 5: Use decimal points instead of commas on the Hg scale.

---

## Referee Comment (RC2) · Anonymous Referee #3 · 11 Sep 2016

General comments: This MS conduct an very interesting study of multiple air pollutants, including Hg, SO2, CO2, CO, NOX emissions through the onboard aircraft measurement in the plume downwind a large coal-fired power plant in Germany, and calculated the emission ratios of Hg versus different air pollutants, and the GOM percentage in the plume. Generally, the work provides a lot of information of the multiple air pollutants emissions. Based on the emission ratios, one can calculate one pollutant emission through the other emissions, these make the pollutant estimation much easier.

Specific comments: (1)Since the air pollutant emissions from the coal fired power plant is largely depended on the boiler type, coal property, and the air pollutant control devices (APCDs), so, the result form one plant maybe differs from the others. Hence, please supplement the information about some basic aspects about the studied power plant, especially the coal property such as the proximate and ultimate analysis (if possible), the configuration of APCDs for NOx, PM and SO2 control.

---

## Author Comment (AC1) · 3 Oct 2016

Response to Anonymous Referee #1

Summary: This manuscript reports measurements of gaseous elemental mercury (GEM), total gaseous mercury (TGM), gaseous oxidized mercury (GOM), carbon monoxide (CO), nitrogen oxides (NOx), sulfur dioxide (SO2), ozone (O3), and meteorological parameters (temperature, pressure, relative humidity, and wind velocity) in the vicinity of Leipzig, Waldhof, and the coal-fired power plant (CFPP) Lippendorf in Germany during August 21 and 22, 2013. The measurements were made from on board a research aircraft and were used to derive Hg/CO, Hg/SO2, and NOx/SO2 emission ratios (ERs) inside the Lippendorf plume. The measurement-based Hg/CO, Hg/SO2, and NOx/SO2 ERs are compared with values calculated from emissions data reported for

the Lippendorf CFPP. The Hg measurements and the Hg/SO2 ERs are further used to estimate GOM/TGM ERs for the Lippendorf CFPP. The main conclusions of the study are that measured Hg/CO and Hg/SO2 ERs are consistent with values reported by the CFPP, whereas observed GOM/TGM ERs are lower than a contemporary global inventory reports. The latter finding is mostly consistent with results of recent field studies. However, one recent study (Landis et al., 2014) reported much higher GOM fractions in a CFPP plume and showed evidence for GOM reduction during downwind plume transport.

Top down verification of Hg emissions from CFPPs is an important research goal. The results and analysis presented by the authors are valuable to better understanding anthropogenic Hg emissions. My main criticisms (detailed under "Specific Comments") are that too little experimental detail is provided on the Hg measurements, in particular the sampling and analysis of GOM. Also, I question the validity/significance of the estimated NO2/NOx ERs. I also feel that further discussion/analysis is needed to relate the authors' GOM/TGM ERs to other values reported in the literature and to the AMAP/UNEP Hg emissions inventory. Additionally, I suggest that the authors seek assistance to correct numerous grammatical errors in the paper. (Some, but not all, grammatical errors are identified below.)

I would recommend that this manuscript be published in ACP after the authors address my "Specific Comments".

We appreciate the comments of referee #1 which helped to improve our manuscript.

Specific Comments:

Page 2 1. Line 1: It is preferable to define compounds when they are introduced (e.g., sulfur dioxide (SO2), carbon monoxide, (CO), etc.). 2. Line 4: (Grammatical error) Insert "The" at the beginning of the sentence starting with "GOM". 3. Line 4: "GOM" should be defined when first used. In general, the authors should check that all acronyms are defined when introduced. 4. Lines 5–6: (Grammatical error) Insert

commas (,) after "were" and "uncertainties". 5. Lines 5–7: (Grammatical error) The sentence beginning with "Measured" is a run-on. This could be corrected by inserting "while" or "whereas" after the last comma. 6. Line 9: Please add some information to qualify the "_10%" value. For example, is this an average or median value? How many individual GOM/TGM ER measurements are represented? If this is an average, the standard deviation, confidence interval, or standard error should be included. 7. Line 16: I suggest you replace "its" with "Hg". 8. Line 24: Insert "Hg" before "emissions". 9. Line 28: Recent work suggests the atmospheric lifetime of elemental Hg may be considerably shorter than 1 yr, though Hg may cycle between elemental and oxidized forms multiple times before being (wet or dry) deposited. I suggest citing a range of lifetimes and adding some more relevant references from the recent literature (e.g., Shah et al., 2016, http://www.atmos-chem-phys.net/16/1511/2016/). 10. Line 29: I suggest rewording the end of the sentence to replace "PM" with "PBM". The acronym "PM" is typically used for "particulate matter". I suggest using "PBM", which is more common in the Hg literature, throughout the manuscript. 11. Line 29: In addition to "washed out" and "rained out" I would include "dry deposited", as this is also an important pathway for GOM (and PBM) removal from the atmosphere.

We assume an undergraduate chemical education of the ACP readers and thus think that chemical formulas do not need additional definition. An exception is NOx which is not a chemical formula but a term coined by air chemists. We now provide its definition as a sum of NO and NO2.

GOM fraction of emissions was determined by three independent methods with three different results. Because of different problems with each of the methods we cannot decide which of the results is less uncertain and, as such, cannot treat them statistically as numbers of the same value. To avoid a long discussion in the abstract we now state that GOM emissions make less than ∼25% of all Hg emissions.

Atmospheric lifetime of elemental mercury: We use the definition of atmospheric lifetime = atmospheric burden/total emission. While atmospheric burden of Hg of some

5600 Mg is quite well known, the estimates of total mercury emissions vary between 11000 Mg (Selin et al. 2008, references in the paper) and 7500 Mg (Pirrone et al., 2010), corresponding to 6 and 9 months, respectively. Estimates based on interhemispheric concentration difference and Junge's formula yields an atmospheric lifetime of ~1 year (Slemr et al., 1985). We give now a range of 6 – 12 months and refer to several references. That does not preclude much shorter local lifetimes such as e.g. during the polar depletion events.

Dry deposition of GOM is now mentioned.

Page 3 12. Line 9: For clarity, I suggest you insert "in-plume" before "GEM oxidation". 13. Lines 12–14: It is not generally true that power plant operators are required to measure and report their emissions. Regulations requiring monitoring and reporting of Hg emissions from CFPPs only recently were enacted in the U.S. I suggest you state that direct measurements of total and speciated Hg emissions from CFPPs are somewhat limited. Then you could state that our understanding of Hg emissions from CFPPs is further limited by the complicating factors you identify (e.g., variable coal composition, complex flue chemistry, etc.). 14. Line 14: "burnt coal" should be "coal burned".

Mercury measurement is mandatory in Germany. This is now mentioned in the text. Proposed EU wide LEVs are now specified.

Page 4 15. Lines 6–8: I suggest you add a reference in support of the claim that Lippendorf is "one of the most modern and efficient CFPPs in Europe". 16. Lines 9–13: I suggest you specify which pollutants were used to classify Lippendorf as the "4th most harmful" and "14th most harmful" emitter. 17. Lines 16–19: Please add some more information to define "LEV". I'm assuming the LEV refers to an in-stack, post-treatment concentration, but it would help if this were clearly stated. 18. Lines 19–20: Emissions monitoring "is mandatory" in Germany or throughout the EU? Many readers will not be familiar with relevant Hg regulations in the study region. Please be more

specific. 19. Line 27: Insert a period (.) at the end of the sentence. 20. Lines 30–31: It seems that once you have defined "normal cruising speed" (i.e., line 25), you don't need to restate the value.

The species used for the rating of Lippendorf as a health damaging emitter are now stated. EU wide LEVs are now mentioned.

Page 5 21. Line 15: Are you referring to the tubing I.D. or O.D.? 22. Line 25: See comment 20.

O.D. is now mentioned in the text.

Page 6 23. Lines 20–31: Too little information is given about how the denuder sampling and analysis was carried out. What was the inlet system that was used for the denuders? The authors should add information to describe the laboratory denuder analysis. How exactly was the analysis performed? Were the denuders loaded in a Tekran 2537/1130 system or was another desorption/analyzer configuration used? Diagrams of both the aircraft sampling system and the laboratory analysis system would be helpful.

Information about denuder sampling and analysis has been added.

Page 7: 24. Line 8: "nitric dioxide" should be "nitrogen dioxide". These compounds should be defined when introduced (see comment 3). 25. Lines 20–29: Too little information is given on calibration and zeroing of the Hg instruments. Were the internal permeation sources verified with a primary source (e.g., a Tekran 2505)? How were the instruments/inlet system zeroed?

Information about calibration and zeroing has been added.

Page 8 26. Lines 10–17: It would help to include a map showing the locations of Lippendorf, Leipzig, and Waldhof together. You could include Hamburg or refer to its direction on the map. It seems it would be helpful to show (or point in the direction of) Leipzig in Fig. 2. 27. Line 11: Remove "a" before "CFPP". 28. Line 27: NOx should

have already been defined earlier.

Map of Germany with Hamburg, Leipzig, Lippendorf and Waldhof has been added.

Page 9 29. Line 9: Insert "of" after "downwind". 30. Lines 27–31: I don't think the discussion is strengthened by discussing the "flight level change" measurements here. The "horizontal flight legs" data seem sufficient to demonstrate the vertical gradient above the 5th flight leg in the PBL.

Page 10 31. Lines 8–10: The fact that the wind direction also points to Lippendorf is also significant.

Page 12 32. Lines 14–16: It is not clear from the text or Table 2 exactly how the Hg/SOÂËĞn2 ERs were calculated from the Tekran Hg measurements. It seems that _Hg/_SO2 was calculated first for each plume encounter and then all values were averaged. If this was the case, why are the number of measurements and the standard deviation (or standard error) not given in Table 2? Some additional details need to be added to Table 2 to explain that the Tekran-based ERs were calculated using the integral method. 33. Line 22: How does the estimated 0.6 to 1.0 NO2/NOx ratio compare to other estimates/observations for similar power plants (c.f., Peischl et al., 2010, http://onlinelibrary.wiley.com/doi/10.1029/2009JD013527/abstract )? A NO/NOx emission ratio of 0.6 would be quite low. You should explain additional assumptions that factor into your estimate. For instance, the observed O3/NOx ratio would seem to be sensitive to the fate of NO2. I question whether your calculation is completely valid or useful.

Tekran instrument is too slow to resolve individual plumes and, therefore, only an integral over all plume crossings can be calculated. The Tekran based Hg/SO2 ERs are thus no averages and as such do not have any standard deviations. This is now stated in the caption of Table 2.

We do not present NO/NOx ratios but NOx/SO2 ERs and O3/NOx enhancement ratios.

[Figure]

The reaction of O3 with NO is very fast and, consequently, the O3/NOx ER should be -1 if all nitrogen oxides were emitted as NO. Table 5 shows that it is roughly the case. We present the O3/NOx ERs as a supporting evidence for the validity of the methods used in this paper.

Page 14 34. Line 8: How did you estimate the thickness of the plume? This should be explained. If you can provide some uncertainty in this number you should be able to estimate uncertainty in the calculated GOM/TGM values, assuming GOM concentrations were in fact negligible outside of the plume. 35. Lines 24–29: Estimated uncertainties should be included with the GOM/TGM percentage values quoted.

The thickness of the plume can be derived directly from the Figure 5. This is now mentioned and the text is modified accordingly.

As mentioned above, GOM fraction was estimated by three independent methods. As such the results cannot be summed up as an average with a corresponding standard deviation. Each method has its specific uncertainties and thus without a reference method we cannot decide which one is more reliable. All results show GOM being less than 25% of total mercury emissions which is consistent with 20% determined by in-stack measurements by Schütze et al. (2015).

Pages 14–15 36. Page 14, Lines 31–32; Page 15, Lines 1–19: It would be helpful to remind the reader at what distances/transport times from the Lippendorf CFPP you encountered the plume. How do these distances/transport times compare with those of the past studies cited? In particular, could differences in sampling (i.e., transport time) reconcile the authors' results with the findings of Landis et al. (2014)? Might the inventory-based GOM fraction still be relevant, but only for very fresh plumes? How representative of the mix of CFPPs represented in the in Wilson et al. (2013) inventory is the Lippendorf CFPP? In other words, might we expect Lippendorf to be a relatively low GOM emitter? A more detailed discussion here would add significant value to the results. Without further consideration of whether the authors' results could

be consistent with those of Landis et al. (2014), or whether Lippendorf is expected to be a relatively low GOM emitter compared to the Wilson et al. (2013) inventory, the conclusions that GOM/TGM is overestimated in the inventories may be misleading. 37. Lines 3–8: See comment 36 (also relevant to the abstract, lines 9–11).

The reduction of GOM into GEM during the transport of the plume could perhaps resolve the difference between our results and those of Landis et al. (2014). But it cannot resolve the difference between GOM measurements directly at the stack: >86% of total mercury emissions measured by Landis et al. (2014) at CFPP Crist and ∼20% reported by Schütze et al. (2015) for CFPP Lippendorf. Based on information by Schütze et al. (2015), the reactions within the FGD system are at least as important for the speciation of the flue gas as the composition of the fuel.

Pages 24 and 29 38. Tables 1 and 6: It isn't clear to me why is the GOM "method lower detection limit" is so much lower than the uncertainty. I suggest you explain how the "method lower detection limit" for GOM was estimated.

The difference is now explained in the text.

Page 32 39. Lines 6–8: I suggest you change "Tekran 1, 2 Hg" in the figure to "GEM, TGM" and eliminate from the caption the explanation of how each Tekran was configured. This explanation was already provided in the text. Page 33 40. Lines 6–8: See comment 38. Page 34 41. Fig. 5: Use decimal points instead of commas on the Hg scale.

Done.

---

## Author Comment (AC2) · 3 Oct 2016

Response to Anonymous Referee #3

General comments: This MS conduct an very interesting study of multiple air pollutants, including Hg, SO2, CO2, CO, NOX emissions through the onboard aircraft measurement in the plume downwind a large coal-fired power plant in Germany, and calculated the emission ratios of Hg versus different air pollutants, and the GOM percentage in the plume. Generally, the work provides a lot of information of the multiple air pollutants emissions. Based on the emission ratios, one can calculate one pollutant emission through the other emissions, these make the pollutant estimation much easier.

Specific comments: (1)Since the air pollutant emissions from the coal fired power plant is largely depended on the boiler type, coal property, and the air pollutant control devices (APCDs), so, the result form one plant maybe differs from the others. Hence, please supplement the information about some basic aspects about the studied power plant, especially the coal property such as the proximate and ultimate analysis (if possible), the configuration of APCDs for NOx, PM and SO2 control.

We were not able to get actual data on the composition of the fuel in 2013, i.e. of lignite and sewage sludge, from the operator of the CFPP Lippendorf. Mercury content of the lignite from two seams of "Vereinigtes Schleenhain" open pit was 0.40 and 0.49 ppm (Rösler et al., 1977), within the range of eastern German lignites of 0.16 – 1.5 ppm (Yudovich and Ketris, 2005). This is now mentioned in the text.

We provide more information about FGD system and refer for details to Schütze et al. (2015). As discussed by Schütze et al. (2015), the chemistry within the FGD system is at least as important as the fuel composition. Schütze et al. (2015) also show a high day-to-day variability of the mercury removal efficiency. Assuming nearly constant FGD operating conditions, this suggests to a large inhomogeneity of the fuel composition.